# Implementing psychosocial interventions for teachers' mental health: Protocol for integrating scoping review with teachers lived experiences in LMICs

**Kanika Malik**(ID)*, **Sadananda Reddy, Yasmine A. Shilla**(ID)**, Aditi Khanna**(ID)**, Ranya R. Kaurr, Abbey Boban**(ID)

Jindal School of Psychology and Counselling, OP Jindal Global University, Sonipat, Haryana, India

\* kmalik1@jgu.edu.in

## Abstract

### Introduction

Teachers are pivotal in shaping educational environments and student development but face significant occupational stress and high rates of mental problems. Despite the availability of various psychosocial interventions, comprehensive evidence of their effectiveness and implementation is limited for this occupational group, especially in low- and middle-income countries (LMICs). This mixed methods study aims to conduct a scoping review of characteristics, effectiveness, and implementation outcomes of psychosocial interventions for teachers' mental health and mental problems, integrating these with teachers' lived experiences to inform the implementation of mental health interventions in LMICs.

### Methods

The study combines a scoping review with insights from an advisory group of teaching professionals with relevant lived experiences (PWLEs). The review will focus on examining psychosocial interventions studies promoting mental health or addressing mental problems among grade 1–12 school teachers of all genders, ages and contexts, except those working in specialized settings, such as special education centers, or disaster- or crisis-stricken zones. Intervention studies exclusively focusing on physical health or job-related outcomes will be excluded. Using pre-defined search terms, quantitative and qualitative research studies, including unpublished literature, will be searched across multiple databases. Titles and abstracts of identified studies will be screened against inclusion criteria, and the potentially relevant sources will be retrieved in full. Data will be extracted using a data extraction sheet developed for the study covering variables related to participant characteristics, intervention characteristics, study context, effectiveness and implementation outcomes.

A purposively selected sample of 10 PWLEs will form the study advisory group and participate in four online workshop-group discussions. The meetings will include a presentation of the scoping review findings, followed by discussions on the relevance of these intervention packages for LMICs, adaptations needed to make them acceptable among school

**Data Availability Statement:** All relevant data are within the paper and its Supporting Information files.

**Funding:** The author(s) received no specific funding for this work.

**Competing interests:** The authors have declared that no competing interests exist.

teachers and feasible for delivery in low-resourced settings like India. The synthesis of the data will employ narrative and thematic approaches to generate actionable insights for implementing psychosocial interventions in LMICs.

## Discussion

This study will provide comprehensive evidence on the characteristics and outcomes of psychosocial interventions for teachers' mental health and mental problems. By integrating insights from teachers with relevant lived experiences, the study will provide practical guidelines for adapting and implementing psychosocial interventions among school teachers in LMICs.

## Trial registration

**Review registration number**: Open Science Framework, doi.org/10.17605/OSF.IO/GF59J.

## Introduction

Teachers play a pivotal role in fostering a healthy school environment and facilitating the holistic development of students [1,2]. To fulfil this role, school teachers must skillfully manage numerous occupational tasks, including catering to students' learning and developmental needs, handling administrative duties, communicating with parents, and supporting students with emotional-behavioral difficulties [3]. The multitude of occupational demands, coupled with limited resources and support at work, makes this one of the most stressful professions [4,5].

Globally, there is a growing concern over the declining mental health of school teachers, characterized by increased rates of depression, anxiety, burnout, and other mental health issues, stemming from the demanding nature of their occupation [6,7]. The repercussions of teachers' poor mental health are far-reaching, contributing to adverse job performance, lower job satisfaction, higher absenteeism and workplace attrition, as well as detrimental psychological and academic development of students with whom these professionals interact [7,8]. The burden is even more significant among teachers in low- and middle-income countries (LMICs), where they often cater to large classrooms; work in settings with limited infrastructure, manpower and other resources; and focus on children with high support needs such as first-generation learners and those with limited financial means [9].

In response to these challenges, various psychosocial intervention programs have been implemented to promote teachers' mental health and alleviate mental problems, such as mental health literacy and first aid programs, mindfulness and yoga interventions, relaxation trainings, and cognitive behavioral therapies [10–14]. A recent systematic review, based on the meta-analysis of 46 control intervention trials, found large to moderate effects of psychosocial intervention on teachers' stress and burnout (g = 0.50–0.93), mental problems (g = 0.38–0.65), and wellbeing (g = 0.38–0.56) [15]. Mindfulness and cognitive behavior interventions were identified as the most frequently used psychosocial approaches to address stress and burnout among teachers [16]. Other reviews that have examined the impact of specific approaches such as mindfulness [17–20] or organizational programs [21], found a weak but positive effect on teachers' mental health, stress and burnout.

However, most of these existing systematic reviews have predominantly concentrated on understanding the intervention impact by focusing on a narrow range of effectiveness studies,

most of which were largely conducted in high-income countries. There has been a paucity of attention towards the acceptability, feasibility and other implementation outcomes, which are important to fully understand the success of an intervention [22,23]. Additionally, there have been limited attempts to include voices from resource-limited LMICs, which account for a large proportion of the global workforce, yet command only a small fraction of structured intervention programs and associated research outputs.

The current mixed methods study aims to bridge this gap by conducting a scoping review of broad range of evidence on characteristics, effectiveness and implementation outcomes of psychosocial interventions for teachers' mental health and mental problems. Additionally, by exploring and incorporating insights from an advisory group of teachers with relevant lived experiences, it aims to identify implications for providing psychosocial interventions for teachers' mental health in LMICs, thereby contributing to a more nuanced understanding of how to support this vital workforce effectively.

For the current study, "characteristics" refers to the features of the psychosocial interventions, including the theoretical framework, settings and mode of delivery, duration, session structure, and provider details. "Effectiveness outcome" refers to the impact of these interventions on reducing symptoms of mental problems and/or improving mental health among teachers. "Implementation outcomes", as defined by Proctor et al [22], will cover acceptability, feasibility, fidelity, adoption, appropriateness, cost, penetration, and sustainability. The "advisory group" refers to purposively selected teaching professionals with relevant lived experiences (PWLEs) who will work with the study team and provide critical insights and feedback on the findings from the scoping review.

## Study question(s)

The study aims to address the following questions:

1. What are the characteristics of psychosocial interventions designed to promote mental health or address mental problems among school teachers?

2. How effective are these psychosocial interventions for enhancing mental health and reducing mental problems among school teachers?

3. What are the implementation outcomes of these psychosocial interventions?

4. What are the practical implications for providing psychosocial interventions for school teachers in LMICs, as identified by discussing the findings of the scoping review with an advisory group of teaching professionals with relevant lived experiences (PWLEs)?

## Methods

The study will use mixed methods approach that combines a scoping review and workshop-group discussions with PWLEs. To achieve the first three objectives, the Preferred Reporting Items for Systematic Reviews and Meta-Analyses Extension for Scoping Reviews guidelines (PRISMA-ScR, [24]) will be used. For the fourth objective, a combination of group discussion and workshop format will be employed. The final protocol was registered prospectively on the Open Science Framework.

### Scoping review

**Eligibility criteria for studies.**   The inclusion and exclusion criteria were developed using the population-concept-context (PCC) framework, which is recommended for scoping

reviews that aim to map key intervention concepts and identify gaps in the literature. The PCC framework is suitable for accommodating a wide range of study designs, aligning with our research objectives [25]. With respect to population (P), the review will consider studies involving in-service school teachers from grades 1 to 12 of any age or gender. Studies that combine teachers and other groups (e.g., students, parents, counselors) will be considered if they analyzed the mental health outcomes for school teachers separately. However, studies focusing on teaching assistants, pre-service teachers, special educators, or educators in higher educational, technical or vocational institutions will be excluded, as their job demands or roles may significantly differ from those of teachers working in school settings.

In terms of concept (C), this review will consider psychosocial intervention studies promoting mental health or addressing mental problems among school teachers. Psychosocial interventions for mental health are defined as individual, group, or organizational level programs that are designed to improve mental well-being, prevent symptoms of psychological distress, and/or modify its determinants (e.g., emotional regulation, critical thinking, mindfulness, self-efficacy). Interventions addressing mental problems are defined as those preventing worsening of subthreshold or diagnosable common mental problems such as depression, anxiety, and post-traumatic stress disorders. Additionally, interventions targeting burnout, which is closely associated with mood symptoms [26], will be also included in the review. Studies exclusively focusing on physical health or physical problems, severe mental disorders, such as psychotic or bipolar disorders, or job-related outcomes (e.g., quality of teaching, teaching efficacy/confidence, or performance in a specific course) will be excluded.

This review will encompass studies conducted in diverse contexts (C) and settings worldwide to ensure a comprehensive understanding of the subject. In line with the study's objective to develop implications for psychosocial interventions in LMICs, we will give greater emphasis in our final report and in discussions with PWLEs to intervention programs conducted in LMICs. We will exclude intervention studies focusing exclusively on teachers operating in special contexts or circumstances, such as natural or man-made disasters, pandemics, humanitarian crises, or specialized education centers. This exclusion criterion is based on the objective to derive insights and recommendations that are applicable and practical for the typical settings in which teachers work.

This scoping review will consider a broad range of intervention studies, utilizing quantitative, qualitative, and/ or mixed methods approach. However, it will exclude systematic or literature reviews of any kind, commentaries, opinion pieces, and prevalence studies

**Search strategy.**   The search strategy aims to locate both published and unpublished (grey) literature available from January 2000 to March 2024. We restricted our search period to ensure that program modes, contexts, and participant experiences are relevant to the current scenario. A preliminary search of Scopus was undertaken to identify relevant articles on the topic. The title, abstract, and keywords sections of these articles were used to develop a comprehensive search strategy. The finalized search strategy includes keywords relating to teachers, psychosocial interventions for mental health promotion and psychosocial interventions for mental problems. The comprehensive search strategy to be used for Scopus database is provided in S1 Appendix, including all search terms, Boolean operators, and limits applied. The search terms and Boolean operators will be adapted appropriately for each database and will be used to retrieve journal articles, book chapters, dissertations, theses, conference papers, unpublished manuscripts, and preprints from the following bibliographic databases: Scopus; ProQuest One Academic, ProQuest Dissertations & Theses Global, ProQuest Central, Education Collection, Education Database, Psychology Database, Publicly Available Content Database, Health & Medical Collection, Education Resources Information Center (ERIC), Nursing & Allied Health Database, Research Library: Social Sciences OR Applied Social Sciences Index

& Abstracts (ASSIA), Social Science Database, Career & Technical Education Database, India Database, East & South Asia Database, Ebook Central, Australia & New Zealand Database, Latin America & Iberia Database, Middle East & Africa Database, UK & Ireland Database, Education Collection (to be accessed through the ProQuest interface); MEDLINE, British Education Index, Cumulative Index to Nursing and Allied Health Literature (CINAHL), Education Abstracts, Educational Administration Abstracts, Teacher Reference Center, eBook Open Access Collection, eBook Collection (to be accessed through the EBSCOhost interface); Global Health, APA PsycArticles Full Text, APA PsycInfo, APA PsycTherapy (to be accessed through the Ovid interface); and ClinicalTrials.gov. Additionally, manual checks will be performed on the reference lists of relevant documents to identify any literature not identified through database searches. Only English language documents (originally written in or officially translated into English) will be included, reflecting the language competencies of the review team.

**Study selection.** Following the search, all identified citations will be collated and uploaded to Rayyan—a free -of-charge, web and mobile application designed for systematic reviews [27]. After the removal of duplicates, the titles and abstracts of the remaining documents will be screened by two independent reviewers according to the inclusion and exclusion criteria laid out. Potentially relevant sources will be retrieved in full, and if necessary, authors will be contacted via email or ResearchGate to obtain the full texts. The full texts will then be assessed in detail against the inclusion criteria by two independent reviewers. Reasons for the exclusion of studies at either the screening or full-text stage will be recorded and reported. Any disagreements that arise between the reviewers at any stage of the selection process will be resolved through discussion or consultation with a third independent reviewer. The selection process will be detailed in the final review, in accordance with the PRISMA flow diagram guidelines [28].

**Data extraction.** Data will be extracted from the selected full texts by two reviewers utilizing a data extraction matrix specifically developed for this review. The data to be extracted include sociodemographic and teaching related information (such as average age, gender distribution, ethnicity, average teaching experience, and teaching level distribution); intervention characteristics (focus, underlying theoretical model, mode and format of delivery, duration, session dosage, type of provider, and provider training); study context and methods (country, design, sample size, comparator type); types of effectiveness outcomes assessed for mental health and/or mental problems (as defined in the concept section); types of implementation outcomes evaluated (as defined by Proctor et al. [22]); and key findings for each of these outcomes, including relevant participant verbatims where applicable (see S2 Appendix for details). The draft data extraction tool will be piloted with a preliminary set of selected documents and modified as necessary. Any further revisions made during the data extraction process will be documented and reported in the final scoping review report. Disagreements arising between the reviewers during the data extraction process will be resolved through discussion.

**Data analysis.** The extracted data will be synthesized using mixed methods approach that integrates narrative synthesis of quantitative data with thematic synthesis of qualitative findings. This combined approach enables a comprehensive understanding of the interventions' effectiveness and the factors affecting their implementation. This approach aligns with the recommended guidelines and previous similar reviews [29,30]. Additionally, a lay summary will be prepared to be discussed with an advisory group of teaching PWLEs.

## Discussions with advisory group of teaching PWLEs

**Design.** The lay summary of the scoping review will be discussed with an advisory group of teaching PWLEs using a combination of workshop and group discussion formats. These

meetings, four in total, will be conducted online using a secure videoconferencing platform such as Zoom. The duration of each meeting will be determined based on the breadth of content to be reviewed and group members' availability.

**Participants and recruitment.**   The advisory group will consist of 10 school teachers, aged 18 and above, who have been teaching for 6 months or more in senior secondary schools in the National Capital Regions of India. The participants must be proficient in written and spoken English, as needed to participate fully in study procedures. The size of the advisory group is in line with previous research using similar integrative reviews [30–32].

These PWLEs will be recruited via social media platforms (e.g., Twitter, Instagram, LinkedIn) and through an outreach team connected to our research group or host institution. Interested individuals will be directed to a secure link providing detailed study information and an expression of interest form. This form will inquire about their sociodemographic information, teaching experience, mental health status, access to internet-enabled devices, and availability for consultation meetings. Upon receiving expressions of interest, study team will purposefully select PWLEs to ensure a diverse representation in terms of age, gender, teaching experience, teaching settings (public/ private schools) and lived experiences of mental problems.

**Procedures.**   Purposively selected participants will receive a digital consent form confirming their participation. At this stage, participants will be given the choice to be acknowledged by name and professional affiliation in the acknowledgments section of any future publications, respecting their preference for recognition and highlighting the value of their contributions. Participants who provide consent will be asked to join online group meetings on selected dates. These meetings will involve presentations on review findings, framed in a language suitable for laypersons, followed by group discussions to gauge advisors' perceptions on the following (i) relevance of the effective and successfully implemented intervention packages, especially those evaluated in LMICs, for the Indian context; (ii) suggested adaptations needed to make these potentially relevant intervention packages more acceptable among teachers; and (iii) the practical implications for delivering these interventions in schools or other suitable locations by peer teachers or other potential providers in LMICs. A topic guide is provided in S3 Appendix. The discussion will be audio-recorded and participants will be provided with gift vouchers upon completion of the study. The amount of each voucher will be decided as per the guidelines of the host institute of the first author.

**Data analysis.**   The recordings will be transcribed and transcripts will be pseudonymized to respect the PWLEs' confidentiality and right to privacy. Once transcribed, transcripts will be cleaned and analysed thematically. The final stage of synthesis will involve consolidating findings from both the scoping review and discussions with PWLEs. We will identify areas of convergence, where both the literature and teaching PWLEs agree, and areas of divergence, where they differ with respect to the potential for implementation and impact. This comprehensive synthesis will inform recommendations for providing psychosocial interventions aimed at improving teachers' mental health in LMICs, specifically in India.

## Discussion

The novelty of this protocol lies in its mixed methods integrative approach, combining a systematic review with lived experience consultations. This scoping review, set to complete by mid-2025, will provide a comprehensive synthesis of evidence on psychosocial interventions for teachers' mental health and mental problems, across various contexts, intervention types and outcomes. By integrating teachers' perspectives with scoping review, we will be able to shed light on unique needs of teachers in LMICs like India and develop guidelines for contextually appropriate interventions to improve teachers mental health, an area often overlooked

in previous reviews on this topic [15–21]. Furthermore, insights from the review and advisory group can guide the development of training programs for counsellors and educators, ensuring they are equipped to deliver these interventions effectively. This integrative approach not only enriches the study findings but also supports patient-centred and participatory research methodologies, recognized for their potential to improve the relevance and impact of research outcomes [33].

However, the study has limitations. Restricting included studies to English may introduce language bias. The decision against appraising evidence quality may affect the conclusions' strength, but it aligns with scoping reviews' aims to map evidence without assessing individual study quality. Additionally, recruitment via social media and the small number of English-speaking advisory PWLEs might cause selection bias, possibly not fully representing the diverse teacher experiences in LMICs. Despite these challenges limiting the generalizability of the findings, the strengths of this study lie in its integration of results from multiple sources to develop a comprehensive set of guidelines for future research on adapting and implementing these interventions for LMICs.

## Supporting information

**S1 Appendix. Search strategy.**
(DOCX)

**S2 Appendix. Data extraction matrix.**
(DOCX)

**S3 Appendix. Topic guide for group discussion with PWLEs.**
(DOCX)

**S4 Appendix. Preferred reporting items for systematic reviews and meta-analyses extension for scoping reviews (PRISMA-ScR) checklist.**
(DOCX)

## Author Contributions

**Conceptualization:** Kanika Malik.

**Methodology:** Kanika Malik, Sadananda Reddy, Aditi Khanna, Ranya R. Kaurr, Abbey Boban.

**Supervision:** Kanika Malik, Yasmine A. Shilla.

**Writing – original draft:** Kanika Malik.

**Writing – review & editing:** Kanika Malik, Sadananda Reddy, Yasmine A. Shilla, Aditi Khanna, Ranya R. Kaurr, Abbey Boban.

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
