## [Decision Letter · Decision Letter 0]

3 Jul 2024

PONE-D-24-12975Implementing Psychosocial Interventions for Teachers' Mental Health: Protocol for Integrating Scoping Review with Teachers Lived Experiences in LMICsPLOS ONE

Dear Dr. MALIK,

Thank you for submitting your manuscript to PLOS ONE. After careful consideration, we feel that it has merit but does not fully meet PLOS ONE’s publication criteria as it currently stands. Therefore, we invite you to submit a revised version of the manuscript that addresses the points raised during the review process. Please note that we have only been able to secure a single reviewer to assess your manuscript. We are issuing a decision on your manuscript at this point to prevent further delays in the evaluation of your manuscript. Please be aware that the editor who handles your revised manuscript might find it necessary to invite additional reviewers to assess this work once the revised manuscript is submitted. However, we will aim to proceed on the basis of this single review if possible. 

We look forward to receiving your revised manuscript.

Kind regards,

Annesha Sil, Ph.D.

Associate Editor

PLOS ONE

Reviewers' comments:

Reviewer's Responses to Questions

**Comments to the Author**

1. Does the manuscript provide a valid rationale for the proposed study, with clearly identified and justified research questions?

Reviewer #1: Yes

2. Is the protocol technically sound and planned in a manner that will lead to a meaningful outcome and allow testing the stated hypotheses?

Reviewer #1: Yes

3. Is the methodology feasible and described in sufficient detail to allow the work to be replicable?

Reviewer #1: Yes

4. Have the authors described where all data underlying the findings will be made available when the study is complete?

Reviewer #1: Yes

5. Is the manuscript presented in an intelligible fashion and written in standard English?

Reviewer #1: Yes

6. Review Comments to the Author

You may also provide optional suggestions and comments to authors that they might find helpful in planning their study.

Reviewer #1: 1) General Comments

The manuscript addresses an important and timely topic: the mental health of teachers and the implementation of psychosocial interventions in low- and middle-income countries (LMICs). The integration of a scoping review with lived experiences is a novel approach that could provide valuable insights. However, the current manuscript has some areas that need clarification. These comments are aimed at improving the clarity, rigour, and overall quality of the manuscript.

Below are detailed comments and suggestions for each section of the manuscript.

2) Abstract

a) The abstract should clearly define the integration of lived experiences. While the current description details the scoping review, it is vague about the timing and conduct of the focus groups and how the data will be integrated. This creates the impression that the discussions with teachers will occur in the future and are not part of the study.

b) The authors refer to the study as a "review" ignoring the fact that this is a mixed-method study that integrates two different methodologies.

c) The authors use the terms "mental health and mental health problems." I suggest using "mental wellbeing and mental health symptoms," as most psychosocial interventions for teachers aim to enhance these aspects. These interventions typically address self-reported symptoms of common mental health issues rather than focusing on diagnosed mental health disorders. Please review these terms throughout the manuscript.

3) Introduction

a) The introduction provides a good background on the importance of teachers' mental health.

4) Review Questions

a) The review questions need to be more specific. Clarify what is meant by "characteristics" and "effectiveness" of interventions, as you did in the third question.

a) The second question includes the impact of these interventions on job performance and other functioning-related outcomes. However, one of the study selection criteria states that job-related outcomes (e.g., quality of teaching, teaching efficacy/confidence, or performance in a specific course) will be excluded. I suggest removing the second portion of the second question.

5) Methods

a) The authors should Include in the context component of the eligibility criteria that only interventions conducted in LMICs will be included.

b) I suggest narrowing the search period to the past 5 to 10 years to ensure relevance.

c) The authors mention that "data to be extracted includes details about the participants" (line 187). I suggest replacing "details" with "sociodemographic information and job-related variables".

d) The data analysis and presentation section is a crucial part of this protocol. The current description does not sufficiently emphasize the mixed methods proposed in this manuscript. The authors should first complete the description of the scoping review portion of the study. Then, in a separate section, they should detail the integration of teachers' suggestions. This new section should thoroughly describe the process of creating lay summaries, the number of focus group sessions, their duration, and the specific roles of teachers in these discussions. For instance, will there be a script for conducting the focus group sessions? What questions will the researchers ask? Additionally, clarify whether teachers participating in the focus group sessions will be compensated for their time.

e) How did the authors determine the number "8 - 10 teachers"? How will the authors ensure equitable and diverse representation of teachers with different backgrounds in this group (e.g. sex, gender, ethnicity, geographic origin, teaching grade/subject/school)?

f) Replace "self-identified mental health concerns" with "lived experiences of mental health symptoms"(line 204). Keep consistency in these terms throughout the manuscript, taking into account that psychosocial interventions do not always target diagnosed mental health disorders, but mental health symptoms.

g) In lines 220 to 223, the authors outline their goals for the discussions with teachers. However, these themes—such as the suitability of interventions, policy changes, and future research—are ambitious and complex for a lay audience. I suggest revisiting this paragraph and adopting a more straightforward script for conducting the focus group sessions.

h) There is a typo in line 226 "scopingreview".

6) Discussion

a) Clearly articulate the practical implications of the scoping review findings for implementing psychosocial interventions in LMICs and emphasize the contribution of integrating teachers' lived experiences to the relevance and applicability of the findings, as the novelty of this protocol.

7. PLOS authors have the option to publish the peer review history of their article (what does this mean?). If published, this will include your full peer review and any attached files.

Reviewer #1: No

---

## [Author Response · Author response to Decision Letter 0]

4 Aug 2024

Reviewer #1: 

1) General Comments

The manuscript addresses an important and timely topic: the mental health of teachers and the implementation of psychosocial interventions in low- and middle-income countries (LMICs). The integration of a scoping review with lived experiences is a novel approach that could provide valuable insights. However, the current manuscript has some areas that need clarification. These comments are aimed at improving the clarity, rigour, and overall quality of the manuscript.

Below are detailed comments and suggestions for each section of the manuscript.

2) Abstract

a) The abstract should clearly define the integration of lived experiences. While the current description details the scoping review, it is vague about the timing and conduct of the focus groups and how the data will be integrated. This creates the impression that the discussions with teachers will occur in the future and are not part of the study.

Response: We have revised the abstract to indicate that the integration of lived experiences will take place following completion of the scoping review. The discussions with PWLEs will consist of four online meetings conducted in a workshop-group discussion format. The process of how these discussions will be conducted is also summarised in the abstract. The revised text is as follows (lines 25-26, 37-43)

The study combines scoping review with insights from an advisory group of teaching professionals with relevant lived experiences (PWLEs) …

A purposively selected sample of 10 PWLEs will form the study advisory group and participate in four online workshop-group discussions. The meetings will include a presentation of the scoping review findings, followed by discussions on the relevance of these intervention packages for LMICs, adaptations needed to make them acceptable among school teachers and feasible for delivery in low-resourced settings like India. The synthesis of the data will employ narrative and thematic approaches to generate actionable insights for implementing psychosocial interventions in LMICs.

b) The authors refer to the study as a "review" ignoring the fact that this is a mixed-method study that integrates two different methodologies.

Response: We have revised the abstract to indicate this study employs dual methodology, integrating a scoping review with discussions involving lived experiences. The revised text is as follows (lines 20-23)

This mixed methods study aims to conduct a scoping review of characteristics, effectiveness, and implementation outcomes of psychosocial interventions for teachers' mental health and mental problems, integrating these with teachers' lived experiences to inform the implementation of mental health interventions in LMICs

Similar changes have also been made in the main text. The revised text is as follows (lines 99-105):

The current mixed methods study aims to bridge this gap by conducting a scoping review of broad range of evidence on characteristics, effectiveness and implementation outcomes of psychosocial interventions for teachers’ mental health and mental problems. Additionally, by exploring and incorporating insights from an advisory group of teachers with relevant lived experiences, it aims to identify implications for providing psychosocial interventions for teachers’ mental health in LMICs, thereby contributing to a more nuanced understanding of how to support this vital workforce effectively..

c) The authors use the terms "mental health and mental health problems." I suggest using "mental wellbeing and mental health symptoms," as most psychosocial interventions for teachers aim to enhance these aspects. These interventions typically address self-reported symptoms of common mental health issues rather than focusing on diagnosed mental health disorders. Please review these terms throughout the manuscript.

Response: We have retained the terms "mental health" and "mental problems" as they better align with the conditions covered in this review. These terms are described in detail in the methods section of the main article. We have also updated the definitions in the Methods sections to make the distinction clearer for the reader (lines 152-161), as follows:

Psychosocial interventions for mental health are defined as individual, group, or organizational level programs that are designed to improve mental well-being, prevent symptoms of psychological distress, and/or modify its determinants (e.g., emotional regulation, critical thinking, problem-solving). Interventions addressing mental problems are defined as those preventing worsening of subthreshold or diagnosable common mental problems such as depression, anxiety, and post-traumatic stress disorders.

3) Introduction

a) The introduction provides a good background on the importance of teachers' mental health.

Response: We appreciate the reviewer’s positive feedback on the clarity of the introduction.

4) Review Questions

a) The review questions need to be more specific. Clarify what is meant by "characteristics" and "effectiveness" of interventions, as you did in the third question.

Response: To address this, we have included definitions of the terms "characteristics" and "effectiveness" after introducing the study objectives. Additionally, we have also included the definitions of “implementation outcomes” and “advisory group.” The revised text is as follows (lines 106-116):

For the current study, "characteristics" refers to the features of the psychosocial interventions, including the theoretical framework, settings and mode of delivery, duration, session structure, and provider details. "Effectiveness" refers to the impact of these interventions on reducing symptoms of mental problems and/or improving mental health among teachers. “Implementation outcomes”, as defined by Proctor et al [22], will cover acceptability, feasibility, fidelity, adoption, appropriateness, cost, penetration, and sustainability. The “advisory group” refers to purposively selected teaching professionals with relevant lived experiences (PWLEs) who will work with the study team and provide critical insights and feedback on the findings from the scoping review.

b) The second question includes the impact of these interventions on job performance and other functioning-related outcomes. However, one of the study selection criteria states that job-related outcomes (e.g., quality of teaching, teaching efficacy/confidence, or performance in a specific course) will be excluded. I suggest removing the second portion of the second question.

Response: To clarify, we intended to exclude articles where the primary focus is on job-related outcomes. However, for the articles where the primary focus is on mental health and mental problems, we were interested in tracking their impact on job outcomes as well. To avoid confusion for the reader and streamline our outputs, we have revised the second review question to remove the impact on job performance and other functioning-related outcomes.

5) Methods

a) The authors should Include in the context component of the eligibility criteria that only interventions conducted in LMICs will be included.

Response: The scoping review aims to cover studies conducted globally to ensure a comprehensive understanding of the subject. Given the limited number of studies from LMICs in previous reviews, we aim to balance this by giving greater weight in our description to intervention programs carried out in LMICs and prioritizing them in discussions with PWLE. Below is the revised version of the manuscript indicating these changes (lines 166-169):

This review will include studies conducted in diverse contexts and settings worldwide to ensure a comprehensive understanding of the subject. In line with the study's objective to develop implications for psychosocial interventions in LMICs, we will give greater emphasis in our final report and in discussions with PWLEs to intervention programs conducted in LMICs.

b) I suggest narrowing the search period to the past 5 to 10 years to ensure relevance.

Response: We have initiated the article screening process, making it challenging to alter the study period at this stage.

c) The authors mention that "data to be extracted includes details about the participants" (line 187). I suggest replacing "details" with "sociodemographic information and job-related variables".

Response: In response to your comment, we have made the following changes to the manuscript (line 226):

The data to be extracted include sociodemographic and teaching related information (such as average age, gender distribution, ethnicity, average teaching experience, and teaching level distribution);

d) The data analysis and presentation section is a crucial part of this protocol. The current description does not sufficiently emphasize the mixed methods proposed in this manuscript. The authors should first complete the description of the scoping review portion of the study. Then, in a separate section, they should detail the integration of teachers' suggestions. This new section should thoroughly describe the process of creating lay summaries, the number of focus group sessions, their duration, and the specific roles of teachers in these discussions. For instance, will there be a script for conducting the focus group sessions? What questions will the researchers ask? Additionally, clarify whether teachers participating in the focus group sessions will be compensated for their time.

Response: We have revised the first paragraph of the methods section to describe the mixed methods approach we are proposing. The revised manuscript text is as follows (lines 135-140):

The study will used a mixed methods approach that combines a scoping review and group discussions with advisory teaching PWLEs. To achieve the first three objectives, the Preferred Reporting Items for Systematic Reviews and Meta-Analyses Extension for Scoping Reviews guidelines (PRISMA-ScR, [26]) will be used. For the fourth objective, a combination of group discussion and workshop format will be employed. The final protocol was registered prospectively on the Open Science Framework.

Additionally, we have created a new subsection that covers the detailed methodology and data analysis related to discussions with PWLEs. The new section describes in detail the process of gathering teachers' feedback, including the number of group meeting, their duration, format, topic guide for group discussions and approach to data analysis synthesis. The updated text is as follows (lines 243-304):

Discussions with advisory group of teaching PWLEs

Design

The lay summary of the scoping review will be discussed with an advisory group of teaching PWLEs using a combination of workshop and group discussion formats. These meetings, four in total, will be conducted online using a secure videoconferencing platform such as Zoom. The duration of each meeting will be determined based on the breadth of content to be reviewed and group members' availability. 

Participants and recruitment

The advisory group will consists of upto 10 school teachers, aged 18 and above, who have been teaching for 6 months or more in senior secondary schools in the National Capital Regions of India. The participants must be proficient in written and spoken English, as needed to participate fully in study procedures. The size of the advisory group is in line with previous research using similar integrative reviews [31-33].

These PWLEs will be recruited via social media platforms (e.g., Twitter, Instagram, LinkedIn) and through an outreach team connected to our research group or host institution. Interested individuals will be directed to a secure link providing detailed study information and an expression of interest form. This form will inquire about their sociodemographic information, teaching experience, mental health status, access to internet-enabled devices, and availability for consultation meetings. Upon receiving expressions of interest, study team will purposefully select PWLEs to ensure a diverse representation in terms of age, gender, teaching experience, teaching settings (public/ private schools) and lived experiences of mental health symptoms.

Procedures

Purposively selected participants will receive a digital consent form confirming their participation. At this stage, participants will be given the choice to be acknowledged by name and professional affiliation in the acknowledgments section of any future publications, respecting their preference for recognition and highlighting the value of their contributions. Participants who provide consent will be asked to join online group meetings on selected dates. These meetings will involve presentations on review findings, framed in a language suitable for laypersons, followed by group discussions to gauge advisors’ perceptions on the following (i) relevance of the effective and successfully implemented intervention packages, especially those evaluated in LMICs, for the Indian context ; (ii) suggested adaptations needed to make these potentially relevant intervention packages more acceptable among teachers ; and (iii) the practical implications for delivering these interventions in schools or other suitable locations by peer teachers or other potential providers in LMICs. A topic guide is provided in S4 Appendix. The discussion will be audio-recorded and participants will be provided with gift vouchers upon completion of the study. The amount of each voucher will be decided as per the guidelines of the host institute of the first author. 

Data analysis

The recordings will be transcribed, and transcripts will be pseudonymized to respect the PWLEs' confidentiality and right to privacy. Once transcribed, transcripts will be cleaned and analysed thematically. The final stage of synthesis will involve consolidating findings from both the scoping review and discussions with PWLEs. We will identify areas of convergence, where both the literature and teaching PWLEs agree, and areas of divergence, where they differ with respect to the potential for implementation and impact. This comprehensive synthesis will inform recommendations for providing psychosocial interventions aimed at improving teachers' mental health in LMICs, specifically in India.

e) How did the authors determine the number "8 - 10 teachers"? How will the authors ensure equitable and diverse representation of teachers with different backgrounds in this group (e.g. sex, gender, ethnicity, geographic origin, teaching grade/subject/school)?

Response: We have revised the manuscript (lines 255-256) to indicate that the ‘size of advisory group is in line with previous research using similar integrative reviews [31-33].’ Further, we will purposively sample from the pool of applicants to ensure diverse representation in terms of age, gender, teaching experience, teaching settings (public/private schools), and lived experiences of mental health symptoms (lines 263-265).

We have acknowledged the small size of advisory group as a limitation in the discussion section (lines 347-352)

Additionally, recruitment via social media and the small number of English-speaking advisory PWLEs might cause selection bias, possibly not fully representing the diverse teacher experiences in LMICs. Despite these challenges limiting the generalizability of the findings, the strengths of this study lie in its integration of results from multiple sources to develop a comprehensive set of guidelines for future research on adapting and implementing these interventions for LMICs.

f) Replace "self-identified mental health concerns" with "lived experiences of mental health symptoms"(line 204). Keep consistency in these terms throughout the manuscript, taking into account that psychosocial interventions do not always target diagnosed mental health disorders, but mental health symptoms.

Response: We have replaced "self-identified mental health concerns" with "lived experiences of mental health problems" (line 265).

g) In lines 220 to 223, the authors outline their goals for the discussions with teachers. However, these themes—such as the suitability of interventions, policy changes, and future research—are ambitious and complex for a lay audience. I suggest revisiting this par

---

## [Decision Letter · Decision Letter 1]

27 Sep 2024

PONE-D-24-12975R1Implementing Psychosocial Interventions for Teachers' Mental Health: Protocol for Integrating Scoping Review with Teachers Lived Experiences in LMICsPLOS ONE

Dear Dr. MALIK,

Thank you for submitting your manuscript to PLOS ONE. After careful consideration, we feel that it has merit but does not fully meet PLOS ONE’s publication criteria as it currently stands. Therefore, we invite you to submit a revised version of the manuscript that addresses the points raised during the review process.

Be sure to address comments raised by reviewer 2

Please submit your revised manuscript by Nov 11 2024 11:59PM If you will need more time than this to complete your revisions, please reply to this message or contact the journal office at plosone@plos.org. Please include the following items when submitting your revised manuscript:A rebuttal letter that responds to each point raised by the academic editor and reviewer(s). You should upload this letter as a separate file labeled 'Response to Reviewers'.A marked-up copy of your manuscript that highlights changes made to the original version. You should upload this as a separate file labeled 'Revised Manuscript with Track Changes'.An unmarked version of your revised paper without tracked changes. You should upload this as a separate file labeled 'Manuscript'.If applicable, we recommend that you deposit your laboratory protocols in protocols.io to enhance the reproducibility of your results. Protocols.io assigns your protocol its own identifier (DOI) so that it can be cited independently in the future. For instructions see: https://journals.plos.org/plosone/s/submission-guidelines#loc-laboratory-protocols. Additionally, PLOS ONE offers an option for publishing peer-reviewed Lab Protocol articles, which describe protocols hosted on protocols.io. Read more information on sharing protocols at https://plos.org/protocols?utm_medium=editorial-email&utm_source=authorletters&utm_campaign=protocols.

We look forward to receiving your revised manuscript.

Kind regards,

Gladys Dzansi, Ph.D, M.Phil, BA, RN

Academic Editor

PLOS ONE

Journal Requirements:

Reviewers' comments:

Reviewer's Responses to Questions

**Comments to the Author**

1. Does the manuscript provide a valid rationale for the proposed study, with clearly identified and justified research questions?

Reviewer #1: Yes

Reviewer #2: Yes

2. Is the protocol technically sound and planned in a manner that will lead to a meaningful outcome and allow testing the stated hypotheses?

Reviewer #1: Yes

Reviewer #2: Yes

3. Is the methodology feasible and described in sufficient detail to allow the work to be replicable?

Reviewer #1: Yes

Reviewer #2: Yes

4. Have the authors described where all data underlying the findings will be made available when the study is complete?

Reviewer #1: Yes

Reviewer #2: Yes

5. Is the manuscript presented in an intelligible fashion and written in standard English?

Reviewer #1: Yes

Reviewer #2: Yes

6. Review Comments to the Author

You may also provide optional suggestions and comments to authors that they might find helpful in planning their study.

Reviewer #1: The authors have thoroughly addressed all the reviewer’s comments and suggestions. No further revisions are needed.

Reviewer #2: Congratulations to the authors for embarking on this interesting study. I have a few comments for the authors to address which hopefully improves the quality and understanding of the entire study.

1. The decision to exclude studies focusing on job-related outcomes (e.g., teaching quality, efficacy) may overlook the interconnectedness between mental health and job performance. Mental health interventions often impact job-related outcomes, and excluding these may lead to an incomplete understanding of the benefits of psychosocial interventions.

2. Given your review topic, it would have been more suitable to use the PICO format (P-population, I-intervention, C-Comparison, O-outcome) to develop your search strategy and eligibility criteria.

3. Present draft of search strategy to be used for at least one electronic database, including planned limits, such that it could be repeated

4. List and define all outcomes for which data will be sought, including prioritization of main and additional outcomes, with rationale.

5. How do the authors mean by this statement: The extracted data will be synthesized using a mixed methods approach, combining narrative and thematic synthesis. Is synthesizing the extracted data using narrative and thematic approach make it a mixed method approach?

7. PLOS authors have the option to publish the peer review history of their article (what does this mean?). If published, this will include your full peer review and any attached files.

Reviewer #1: No

Reviewer #2: No

---

## [Author Response · Author response to Decision Letter 1]

26 Oct 2024

Dear Editor and Reviewers,

We sincerely appreciate the time and effort you have invested in reviewing the manuscript. We have carefully considered each point raised and have made the necessary revisions to address your concerns. Our responses and modification to the manuscript are described below, with the page and line numbers referring to tracked version:

Reviewer #1:

The authors have thoroughly addressed all the reviewer’s comments and suggestions. No further revisions are needed.

Response:

Thank you for your positive feedback. We are pleased that our revisions have met your expectations and that you find the manuscript satisfactory.

Reviewer #2:

Congratulations to the authors for embarking on this interesting study. I have a few comments for the authors to address which hopefully improve the quality and understanding of the entire study.

Comment 1:

The decision to exclude studies focusing on job-related outcomes (e.g., teaching quality, efficacy) may overlook the interconnectedness between mental health and job performance. Mental health interventions often impact job-related outcomes, and excluding these may lead to an incomplete understanding of the benefits of psychosocial interventions.

Response:

Thank you for your comment. While we acknowledge that mental health and job performance are interconnected, this review focuses specifically on mental health outcomes to provide a detailed analysis of psychosocial interventions in this domain. The impact on job-related outcomes, although important, is beyond the scope of this review but represents a valuable area for future research. We will keep this in mind for future secondary analyses that may arise from this work, where we can delve deeper into the impact of psychosocial interventions designed for mental health on job-related and other functional outcomes.

Comment 2:

Given your review topic, it would have been more suitable to use the PICO format (P-population, I-intervention, C-comparison, O-outcome) to develop your search strategy and eligibility criteria.

Response:

We appreciate your suggestion regarding the use of the PICO framework. In designing our scoping review, we considered the pros and cons of both the PICO and PCC (Population-Concept-Context) frameworks.

Given that our review includes a diverse range of study designs—including quantitative, qualitative, and mixed-methods studies—and aims to map the breadth of literature on psychosocial interventions in mental health and problems, we determined that the PCC framework was more appropriate. The PCC framework is recommended for scoping reviews by the Joanna Briggs Institute and allows for broader inclusion criteria, which aligns with our objective to identify gaps and trends in the existing research.

We have updated the Methods section to include the rationale for choosing PCC framework, as below (page 8, lines 130-133):

The inclusion and exclusion criteria were developed using the population-concept-context (PCC) framework, which is recommended for scoping reviews that aims to map key intervention concepts and identify gaps in the literature. The PCC framework is suitable for accommodating a wide range of study designs and does not require a comparison group, which aligns with our research objectives [27]. 

We hope this explanation clarifies our choice of framework and demonstrates its suitability for our study.

Comment 3:

Present draft of search strategy to be used for at least one electronic database, including planned limits, such that it could be repeated.

Response:

To facilitate transparency and reproducibility of the review process, the full search strategy for the Scopus database is already included in Supplementary Appendix S1. This includes search terms and keywords used, Boolean operators applied, filters and limits set, date range of the search

We have also updated the Methods section to make it explicit the location of the sample strategy and what it contains, as given below (page 10, lines 174-175):

The comprehensive search strategy to be used for Scopus database is provided in S1 Appendix, including all search terms, Boolean operators, and limits applied.

Please let us know if further details are required or if there were specific aspects you would like us to expand upon.

Comment 4:

List and define all outcomes for which data will be sought, including prioritization of main and additional outcomes, with rationale.

Response:

In our review, the effectiveness outcome data will be extracted for variables related to mental health and/or mental problems, and implementation outcome data will be extracted for domains such as acceptability, feasibility, fidelity and others as proposed by Proctor et al (2011). We have revised the Data extraction section of manuscript and S2 Appendix to include details of outcomes as follows:

Manuscript (page 12, lines 216-219):

…types of effectiveness outcomes assessed for mental health and/or mental problems (as defined in the concept section); types of implementation outcomes evaluated (as defined by Proctor et al. [22]); and key findings for each of these outcomes, including relevant participant verbatims where applicable (see S2 Appendix for details). 

S2 Appendix (supporting information document, line 20)

Type of effectiveness outcome assessed for mental health (e.g., well-being, psychological distress, and/or determinants of mental health such as mindfulness, emotional regulation, critical thinking, self-efficacy), and/ or mental problems (such as depression, anxiety, burnout, and post-traumatic stress disorder), quantitative and/or qualitative outcome measure used, key findings on measure, relevant participant verbatims

Type of implementation outcome assessed (i.e. acceptability, feasibility, fidelity, cost, adoption, appropriateness, penetration), quantitative and/or qualitative outcome measure used, key findings on measure, relevant participant verbatims

Comment 5:

How do the authors mean by this statement: "The extracted data will be synthesized using a mixed methods approach, combining narrative and thematic synthesis." Is synthesizing the extracted data using narrative and thematic approach make it a mixed method approach?

Response:

Thank you for seeking clarification. In our scoping review, we define a mixed methods synthesis as the integration of both quantitative and qualitative data to provide a comprehensive analysis of the research topic.

• Narrative Synthesis: This approach involves summarizing and explaining the findings of multiple quantitative studies in a descriptive manner. It allows us to identify patterns, relationships, and trends across the quantitative data extracted, such as intervention effectiveness and measured outcomes.

• Thematic Synthesis: This method is used to analyze qualitative data from studies that include qualitative findings or components. Through thematic analysis, we can identify common themes, perceptions, and experiences related to the interventions.

By combining these two synthesis methods, we effectively integrate quantitative results with qualitative insights, which enriches our understanding of both the measurable effects and the contextual factors influencing the interventions.

We have revised the Data analysis section to clarify this, as below (page 12, lines 227-230):

The extracted data will be synthesized using mixed methods approach that integrates narrative synthesis of quantitative data with thematic synthesis of qualitative findings. This combined approach enables a comprehensive understanding of the interventions' effectiveness and factors affecting their implementation. 

We trust that this explanation clarifies how the use of narrative and thematic synthesis constitutes a mixed methods approach in our review.

---

## [Decision Letter · Decision Letter 2]

26 Dec 2024

Implementing Psychosocial Interventions for Teachers' Mental Health: Protocol for Integrating Scoping Review with Teachers Lived Experiences in LMICs

PONE-D-24-12975R2

Dear Dr. MALIK,

We’re pleased to inform you that your manuscript has been judged scientifically suitable for publication and will be formally accepted for publication once it meets all outstanding technical requirements.

Kind regards,

Muhammad Shahzad Aslam, Ph.D.,M.Phil., Pharm-D

Academic Editor

PLOS ONE

Additional Editor Comments (optional):

Reviewers' comments:

Reviewer's Responses to Questions

**Comments to the Author**

1. Does the manuscript provide a valid rationale for the proposed study, with clearly identified and justified research questions?

Reviewer #2: Yes

2. Is the protocol technically sound and planned in a manner that will lead to a meaningful outcome and allow testing the stated hypotheses?

Reviewer #2: Yes

3. Is the methodology feasible and described in sufficient detail to allow the work to be replicable?

Reviewer #2: Yes

4. Have the authors described where all data underlying the findings will be made available when the study is complete?

Reviewer #2: Yes

5. Is the manuscript presented in an intelligible fashion and written in standard English?

Reviewer #2: Yes

6. Review Comments to the Author

You may also provide optional suggestions and comments to authors that they might find helpful in planning their study.

Reviewer #2: The authors have done a good job in addressing all the reviewer's comments and strengthening the quality and clarity of the manuscript.

7. PLOS authors have the option to publish the peer review history of their article (what does this mean?). If published, this will include your full peer review and any attached files.

Reviewer #2: No

---

## [Editor Report · Acceptance letter]

17 Jan 2025

PONE-D-24-12975R2 

PLOS ONE

Dear Dr. MALIK, 

I'm pleased to inform you that your manuscript has been deemed suitable for publication in PLOS ONE. Congratulations! Your manuscript is now being handed over to our production team.

Kind regards, 

on behalf of

Dr. Muhammad Shahzad Aslam 

Academic Editor

PLOS ONE